# Tri-Model Therapy: Combining Macrocyclic Lactone, Piperazine Derivative and Herbal Preparation in Treating Humpsore in Cattle

**DOI:** 10.3390/vetsci8020027

**Published:** 2021-02-13

**Authors:** Perumal Ponraj, Arun Kumar De, Samiran Mondal, Sanjay Kumar Ravi, Sneha Sawhney, Gopal Sarkar, Asit Kumar Bera, Dhruba Malakar, Ashish Kumar, Laishram Brojendra Singh, Sheikh Zamir Ahmed, Kangayan Muniswamy, Bosco Augustine Jerard, Debasis Bhattacharya

**Affiliations:** 1Animal Science Division, ICAR-Central Island Agricultural Research Institute, Port Blair 744101, Andaman and Nicobar Islands, India; perumalponraj@gmail.com (P.P.); skravivet@gmail.com (S.K.R.); snehasawhney88@gmail.com (S.S.); swamy02_vet@yahoo.co.in (K.M.); debasis63@rediffmail.com (D.B.); 2Department of Veterinary Pathology, West Bengal University of Animal and Fishery Sciences, Kolkata 700037, West Bengal, India; vetsamiran@gmail.com (S.M.); gsarkar999@gmail.com (G.S.); 3Reservoir and Wetland Fisheries Division, ICAR-Central Inland Fishery Research Institute, Barrackpore, Kolkata 700120, West Bengal, India; asitmed2000@yahoo.com; 4Animal Biotechnology Centre, National Dairy Research Institute, Karnal 132001, Haryana, India; dhrubamalakar@gmail.com; 5CTARA, IIT Bombay, Mumbai 400076, Maharashtra, India; ashish07.vet@gmail.com; 6Krishi Vigyan Kendra, ICAR-Central Island Agricultural Research Institute, Port Blair 744101, Andaman and Nicobar Islands, India; lbrojendra@gmail.com; 7Social Science Section, ICAR-Central Island Agricultural Research Institute, Port Blair 744101, Andaman and Nicobar Islands, India; zamir562.za@gmail.com; 8Horticulture and Forestry Division, ICAR-Central Island Agricultural Research Institute, Port Blair 744101, Andaman and Nicobar Islands, India; jerardba@gmail.com

**Keywords:** humpsore, tri-model therapy, antioxidant, oxidative stress profiles, cortisol, histopathology

## Abstract

Stephanofilariasis or humpsore is a chronic parasitic dermatitis of cattle. Various treatment regimens were attempted in the past but were found to be partially effective. Here, we claim a successful treatment regime using an FDA-approved macrocyclic lactone, a piperazine derivative, and an herbal preparation. Twenty-four cattle (18 affected and 6 unaffected) were selected and divided into Gr 1: positive control (infected without treatment; *n* = 6), Gr 2: treatment group (infected with treatment with ivermectin; *n* = 6), Gr 3: treatment group (infected with treatment with tri-model therapy including ivermectin, diethylcarbamazine citrate, and an herbal ointment, *n* = 6), and Gr 4: negative control (non-infected animals; *n* = 6). In Gr 2 and Gr 3, treatment to the ailing animals were given for 30 days. Lesion was significantly reduced in day 15 of post-treatment and completely healed on day 30 of post-treatment in Gr 3. Tri-model therapy recorded significant improvement in the surface area of the sore as compared to ivermectin administration alone. Antioxidants were increased and malondialdehyde (MDA) and cortisol concentrations were decreased significantly (*p* < 0.05) in Gr 3 than in untreated control group at day 14, 21 and 28. Histopathological changes in infected animals were characterized by parakeratotic hyperkeratosis along with presence of nucleated keratinocytes. There were infiltrations of polymorphonuclear cells specially eosinophils along with a few monomorphonuclear cells. Microfilarial organism was observed beneath the epidermis, which was surrounded by fibrocytes and infiltrated cells. In the tri-model-treated animal after recovery, the skin revived a normal architecture. Therefore, tri-model therapy has the potential to cure humpsore.

## 1. Introduction

Stephanofilariasis popularly known as humpsore, caused by the filarial worm *Stephanofilaria assamensis* and transmitted by *Haematobia irritans*, *Musca conducens*, *Musca planiceps,* and *Musca autumnalis*, is a common chronic skin disorder of cattle endemic to hot humid climate of Indian subcontinent [1,2,3,4]. Stephanofilariasis is responsible for damage of hair follicles and causes chronic dermatitis in cattle. Humpsore causes significant economic losses to the dairy farmers as the market value of the animals reduces due to damaged skin [5]. The typical lesions occur mostly in and around hump (92.60%) and rarely in udder, sternal area, and other parts of the body [6]. The size of sore varies from a few cm to more than 30 cm [6]. Stephanofilariasis shows higher prevalence rate in exotic and its crossbred (20.17%) as compared to zebu cattle (16.14%) [7], mostly prevalent in dark-colored cattle [8] and dry season of the year due to abundance of dipteran population [2,3,4]. Different workers both in India and other parts of the world attempted to treat the disease with partial success rate. The treatment regimens include surgical intervention, use of cauterizing agents [9], external application of petroleum jelly [10], subcutaneous administration of Antimosan with application of 1% gentian violet [11], parenteral administration of 8% Trichlorphon [12], combination of antimony potassium tartrate and 4–8% phenothiazine ointment [13], tobacco ointment alone [13], injectable preparation of levamisole hydrochloride with zinc oxide ointment [2], combination of ivermectin, levamisole, and Mastilep ointment [14,15], ivermectin and Topicure spray [16], and ivermectin and topical application of zinc oxide ointment [17]. Till date, a very effective treatment regime for the disease is not available, and we, here, propose a “tri-model therapy” combining a macrocyclic lactone (ivermectin), a piperazine derivative (DECC), and an herbal preparation (Himax ointment), which was found very effective in treating the disease.

Ivermectin, a macrolide anthelmintic, is very effective against microfilaria and has less effect on adult filarial worm, whereas diethylcarbamazine (DECC) is used for prophylaxis against filarial worm in canine [18]. Therefore, in the present study, we have combined these two FDA-approved drugs along with Himax ointment as antibacterial and fly repellent with the hypothesis that they will be effective as chemotherapeutic and chemoprophylactic agents against stephanofilariasis in cattle. Moreover, the effect of the therapy on antioxidant and oxidative stress profiles and skin histopathological alterations was evaluated to judge the efficacy of our proposed tri-model therapy.

## 2. Materials and Methods

### 2.1. Ethics Approval

Institute Animal Ethics Committee (IAEC) of ICAR-Central Island Agricultural Research Institute (ICAR-CIARI), Port Blair, Andaman and Nicobar Islands, India, approved the study and all the methods were performed in accordance with the relevant national guidelines and regulations. The ethic code of the study is “F. No. NB. A & NI. RO. IAEC/234/RO-206 (4)/2019-2020 dated 3 December 2019.”

### 2.2. Therapeutic Agents

Ivermectin injection (HITEK^TM^) was purchased from KachhelaMedex Pvt. Ltd., Nagpur, India. Diethylcarbamazine citrate (DECC) (Hetrazan) and Himax ointment were procured from Wyeth India, Mumbai, India, and Indian Herbs Overseas, Uttar Pradesh, India, respectively.

### 2.3. Animals and Experimental Design

The present study was conducted at the Cattle Breeding Farm, ICAR-Central Island Agricultural Research Institute, Port Blair, Andaman and Nicobar Islands, India (11.6060° N, 92.7058° E) from December 2019 to January, 2020 (dry season). The experiment was conducted on crossbred cattle (Indigenous and Jersey). Animals with a single circular or irregular-shaped large/chronic wound (>12 cm diameter) were selected for the current experiment. Humpsore was confirmed by detection of microfilaria in skin scraping (Figure 1) of the infected animals. The animals were 4–6 years old with body condition score of 4–5 out of 10. A total of twenty-four cattle were chosen and were randomly divided into four groups, namely, Gr 1: positive control, Gr 2: treatment group with ivermectin alone, Gr 3: treatment group with tri-model therapy and Gr 4: negative control, with six animals in each group. Experimental animals were maintained under uniform management with semi-intensive system of rearing where they were allowed for grazing from 7:00 a.m. to 12:00 p.m. Feeding and watering was done as per the farm schedule. *Ad libitum* clean drinking water was available. The details of the groups and treatment protocol are as follows.

Group 1: Positive control (infected animals without treatment)

Group 2: Treatment (infected animals with treatment with ivermectin alone); Humpsore was cleaned with a mild liquid soap (Lifebuoy liquid soap, Hindustan Unilever Ltd., Mumbai, India) and crusts were removed. Papaya (*Carica papaya*) mist was applied on the wound for about 30 min on first day. Injection of ivermectin @ 200 µg/kg body weight was administered two times by subcutaneous route at 15 days interval (day 0 and 15).

Group 3: Treatment (infected animals with treatment with tri-model therapy); Humpsore was cleaned with a mild liquid soap (Lifebuoy liquid soap, Hindustan Unilever Ltd., Mumbai, India) and crusts were removed. Papaya (*Carica papaya*) mist was applied on the wound for about 30 min on first day. Herbal antiseptic and fly-repellent ointment (Himax ointment containing *Curcuma longa* rhizome extract 0.40 g, *Ponggamia pinnata* seed oil 4.55 g, *Cedrus deodara* wood oil, and Gandhaka powder 0.50 g per 100 g of ointment) was applied daily on the wound for 30 days. Injection of ivermectin @ 200 µg/kg body weight was administered two times by subcutaneous route at 15 days interval (day 0 and 15). Diethylcarbamazine citrate 6 mg/kg body weight was administered orally daily for 30 days.

Group 4: Negative control (non-infected animals).

### 2.4. Measurement of the Wound

Wound on each animal was measured on day 0, day 15, and day 30 post-treatment. We used a disposable paper ruler to measure the length and width of the wound. Surface area of the wound was measured by multiplying greatest length head-to-toe and greatest width perpendicular to length [19].

### 2.5. Collection of Blood and Estimation of Antioxidant and Oxidative Stress Profiles

Blood samples were collected by venipuncture of jugular vein into heparin tubes (20 IU of heparin/mL of blood) at day 0 and weekly interval thereafter (day 7, 14, 21, and 28). The blood samples were centrifuged at 1200× *g* for 15 min at 4 °C. The plasma samples were separated, labeled, and preserved at −80 °C for further analysis. Total antioxidant capacity, superoxide dismutase, and catalase in the blood plasma were estimated using commercial kits (Cayman Chemical Company, Ann Arbor, MI, USA) as per the manufacturer’s guidelines. Malondialdehyde in plasma was estimated as per the method described by Shah and Walker [20] using TBA (Thiobarbituric acid)—TCA (Trichloro acetic acid) method. Blood cortisol level was estimated using a commercially available ELISA kit (Cayman Chemical Company, Ann Arbor, MI, USA). All these antioxidants, oxidative free radical, and cortisol were estimated in a 96-well clear polypropylene microplate using a Microplate Reader (Alere Medical Pvt Ltd., Gurugram, India, AM 2100).

### 2.6. Histopathological Examination

Skin biopsy of humpsore-affected cattle was collected before and after the treatment. The collected samples were preserved at 10% formalin solution and subsequently processed, embedded with paraffin, sectioned, and stained with hematoxylin and eosin method.

### 2.7. Statistical Analysis

Data were presented as the mean ± SEM. Data used in the study were tested for normality before analysis using Shapiro–Wilk statistics. Data present a normal distribution and were homoscedastic. Means were analyzed by two-way analysis of variance (ANOVA), followed by the Tukey’s post hoc test to determine significant differences among the different experimental groups and among days of sample collection using the SAS/PC computer program (Statistical Analysis System for Windows, SAS Version 9.3; SAS Institute, Inc., Cary, NC, USA, 2001). Differences with values of *p* < 0.05 were considered to be statistically significant.

## 3. Results

### 3.1. Clinical Examination

During the study period of 30 days, the animals thatdid not receive treatment had the humpsore lesions. In Gr 2, the average surface area (mean ± SEM) of the sore (cm^2^) reduced from 171.83 ± 11.68 to 100.04 ± 7.31 on day 15 and to 50.86 ± 3.17 on day 30. In Gr 3, the average surface area (mean ± SEM) of the sore (cm^2^) reduced from 174.94 ± 13.73 to 31.78 ± 2.97 on day 15 and to 1.49 ± 0.19 on day 30. Moreover, significantly reduced sore surface area in Gr 3 on day 15 and day 30 as compared to Gr 2 was observed (Figure 2). In Gr 3, no oozing from the wound and complete healing of wound was observed in 30 days of post-treatment (Figure 3).

### 3.2. Antioxidant and Oxidative Stress Profiles

Results of the antioxidant and oxidative stress profiles revealed that TAC, SOD, and CAT were lower and MDA was higher significantly (*p* < 0.05) in humpsore-affected groups (Gr 1, Gr 2, and Gr 3) than those of unaffected negative control group (Gr 4) at day 0. The levels of these antioxidants were increased, and oxidative stress profile was reduced significantly (*p* < 0.05) in treated groups (Gr 2 and Gr 3) than in untreated group (Gr 1) at day 14, 21, and 28 except for SOD in day 14 where no significant difference between Gr 2 and Gr 1 was observed. Moreover, when the two treated groups were considered, significantly increased values of TAC, SOD, and CAT and decreased values of MDA were observed in Gr 3 as compared to those of Gr 2 on day 14 onwards, except for SOD on day 21 where no significant difference between the two groups was observed. When the values at different time points in a particular group was considered, these antioxidants were increased and oxidative stress profile was reduced significantly from day 0 to day 28 in Gr 2 and Gr 3, whereas in Gr 1 and Gr 4, no significant variation was observed throughout the study period (Figure 4). Similarly, level of cortisol was decreased in treated groups (Gr 2 and Gr 3) than untreated group (Gr 1) at day 14, 21, and 28 (Figure 5). Moreover, decreased cortisol values were detected in Gr 3 as compared to Gr 2 at day 7, 14, 21, and 28. TAC showed positively significant correlation with SOD (*p* < 0.05; r = 0.88) and CAT (*p* < 0.05; r = 0.96), whereas negative significant correlation with MDA (*p* < 0.05; r = 0.84) and cortisol (*p* < 0.05; r = 0.95) in Gr 3 was observed.

### 3.3. Histopathology

The microscopic lesion in the untreated animal consisted of hyperkeratosis, i.e., thickening of stratum corneum of the epidermis, which was the outermost layer of the skin. The outermost layer of the skin consisted of flat cells keratinocytes, an epithelial cell that differentiated to produce keratin. In the present investigation, nucleus was present and the cytoplasms of those epithelial cells were replaced by keratin, thought to be derived from tonofibril of the deeper layer of the epidermis. Hyperkeratosis was parakeratotic in nature because keratinocytes retained their nucleus. Both stratum spinosum and stratum granulosum were proliferated. Cytoplasm of cells of stratum granulosum layers contained granules of keratohyalin, which appeared to be involved in the process of formation of soft keratin. The rete pegs were elongated and hyperplastic, which led to acanthosis. There were infiltrations of polymorphonuclear cells specially eosinophils along with some monomorphonuclear cells. Infiltrated cell formed microabscess containing debris within. Dermis showed extensive reaction causing dermatitis. Microfilarial organism was observed beneath the epidermis, which was surrounded by some fibrocytes and infiltrated cells (Figure 6). There was no presence of hair follicle in the biopsy sample. In the tri-model-treated animal (Gr 3), there was almost normal epidermis with different layers of cells. There was no infiltration, and normal adnexal structures were present in the biopsy material (Figure 7).

## 4. Discussion

Humpsore in cattle is a common chronic skin disease and the disease is highly prevalent in hot humid climate of Indian subcontinent and Southeast Asian countries [2]. It causes pruritus and damage to the hair follicles. Depending on the stage of infection, it causes ulcerative nodular dermatitis, exudation, granulation, ulceration, and incrustation. The farmers suffer severe economic losses as the affected males are unsuitable for draught purposes and affected females lose their productivity, growth rate, and fertility rate [1]. It is also associated with unclean or unhygienic milk production [16]. Therefore, development of a very successful treatment therapy is very important to eradicate the disease.

Various workers both in India and in other parts of the world suggested different treatment protocols for eradication of the disease but till date, a very successful therapy has yet to be developed. A single dose of levamisole HCl @ 1 mL (182 mg) per 15–20 kg body weight administered either subcutaneously or intramuscularly along with topical application of an ointment containing 10% zine oxide and 1% resorcinol showed efficacy ranging from 65.8% to 97.2% [3]. Several other treatment protocols including use of cauterizing agents [9], external application of petroleum jelly [10], subcutaneous administration of Antimosan with application of 1% gentian violet [11], parenteral administration of 8% Trichlorphon [12], combination of antimony potassium tartrate and 4–8% phenothiazine ointment [13], and tobacco ointment [13] have been tried with varied success rate. Some other drugs have been tried to treat filarial diseases like melarsomine is used as the drug of choice to treat heartworm [21]. In recent times, most of the treatment protocols cantered around injection of ivermectin either alone or combination with different ointments [13,15,16]. In a study by Islam et al. [16], ivermectin therapy showed significant clinical improvement of humpsore in 90% cases. In the present study, we have developed tri-model therapy combining ivermectin, diethylcarbamazine citrate, and anherbal preparation, which was found more effective than a standard regimen (ivermectin alone). Tri-model therapy recorded significant reduction in the surface area of the sore as compared to ivermectin administration at the end of 30 days of treatment. The efficacy of the therapeutic approach was assessed on the basis of sore healing. All the animals treated with tri-model therapy cured completely after the treatment period and no microfilaria was detected in the skin scraping of treated animals. Moreover, recurrence of the disease was not detected in any of the treated animals (tri-model therapy group) for 1 year. However, further study is needed to record the recurrence rate over a longer time interval. Ivermectin though very effective against microfilaria has less effect on adult filarial worm, whereas diethylcarbamazine kills both microfilariae and adult worms [22]. Therefore, in the present study, we combined there two drugs with the hypothesis that they would be an ideal treatment option for humpsore. In addition, we have included Himax ointment as antibacterial and fly-repellent agent.

Ivermectin causes paralysis and death of microfilaria by targeting the postsynaptic glutamate-gated chloride channel receptor (GluClR) [23]. Moreover, ivermectin interferes with the ability of microfilariae to evade the immune system, resulting in the host’s own immune response being able to overcome the immature worms and thus kills those microfilariae [24]. It is well known that ivermectin administered subcutaneously persists for 10–21 days [25,26,27]. Therefore, we administered ivermectin at the interval of 15 days (day 0, 15) in the study to treat humpsore. DECC acts on filarial worm in two different ways; first, by decreasing muscular activity of the nematode and second, by altering the microfilarial surface membranes making them more susceptible to host defenses [28]. DECC is well absorbed following an oral administration and reaches all parts of the body within 25 min after its intake with peak plasma concentration reaches within 1–2 h. The plasma half-life varies from 6.1 to 8.1 h. The standard DECC treatment regime is 6 mg/kg per day over a 10–20-day period in filariasis treatment [29]. Therefore, in the present study, oral DECC therapy at 24 h interval was used to treat humpsore.

Hydrophilic, lipophilic, and enzymatic antioxidants are significantly higher in epidermis than in dermis [30]. Normal bacterial flora in the skin generates reactive oxygen species (ROS). Mechanical injury, abnormal humidity, immunodeficiency, or metabolic disorders can result in increased bacterial infection, which may lead to increased bacterial ROS production. This could act synergistically with ROS from host phagocyte cells to increase the generation of pro-inflammatory agents causing tissue damage [31]. ROS are centrally involved in all wound healing processes as low concentrations of ROS generation are required to fight against invading microorganisms and cell surviving signaling [32]. In line, antioxidant and anti-inflammatory properties of several antioxidant strategies have proven beneficial to improve non-healing state [33]. Ongoing oxidative stress, associated with lipid peroxidation, protein modification, and DNA damage has been shown to impair wound healing processes via increased cell apoptosis and senescence [34]. Clinical studies suggest that non-healing wounds are maintained in highly oxidizing environment, which lead to impaired wound repair. Clinical condition such as tissue hypoxia due to occultation of the worm on the passage of the vessels is typically associated with highly oxidizing environments. Therefore, in the present study, significantly higher level of oxidative radical was observed in the humpsore-affected cattle than in normal unaffected animals.

Normally during wound healing process, concentration of tissue enzymatic and non-enzymatic antioxidants is reduced as these antioxidants are needed for healing processes [35]. It is not possible to supply antioxidants due to aging asthe process of wound healing leads to increased production of ROS, which further impose the animals to be vulnerableto oxidative stress [36]. It would appear that a physiologically significant contribution to lipophilic antioxidants of skin layers and surface is derived from sebaceous gland secretion [37]. Similarly, in the present study, result revealed that the unaffected animals have significantly higher level of antioxidants than the affected animal groups. In the affected animals, the treatment has enhanced the healing process and improved the level of antioxidants and decreased the oxidative stress profile.

Healing of the epidermis normally involves migration of keratinocytes from the wound edge which undergo proliferation, differentiation, and apoptosis. Histopathological studies revealed presence of hyperproliferative stem cells and actively proliferating keratinocytes at the ulcerated wound margin [38]. Despite the presence of hyperproliferative keratinocytes, healing of the ulcerated wound was slow, suggesting that the problems actually lie with distorted organization of the wound bed. In these animals, it may have been caused by infection and impaired nutritional supply, which impairs keratinocytes migration. ROS may be implicated in impairment of early stages of healing in the wound bed in ulcerated wounds through inflammatory process, with an imbalance of ROS and antioxidant process. Therefore, in the present study, we observed that the humpsore wound healing was quicker and faster in the tri-model-treated group than ivermectin-treated group.

## 5. Conclusions

It may be concluded that tri-model therapy combining a macrocyclic lactone, a piperazine derivative, and an herbal preparation has the potential to cure humpsore lesions. The treatment we propose represents an improvement compared to the already established treatments. Moreover, it improves the antioxidant level and reduces oxidative stress profiles and cortisol concentration.

## Figures and Tables

**Figure 1 vetsci-08-00027-f001:**
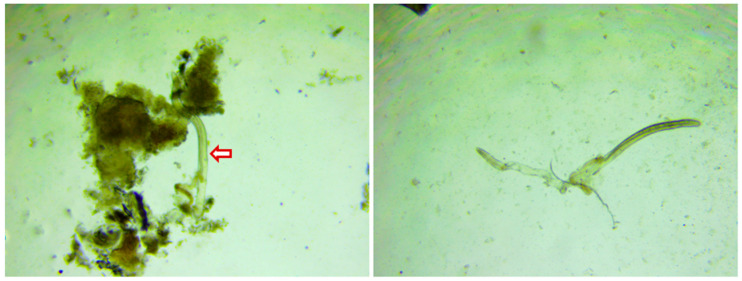
Microfilaria in humpsore-infected skin scraping (40×). Pinch of skin from the infected animal was collected in phosphate buffered saline (PBS) and transported to the laboratory. The skin scraping was minced in PBS and concentrated by centrifugation (250× *g*) for 15 min and sediment was examined under 40× magnification.

**Figure 2 vetsci-08-00027-f002:**
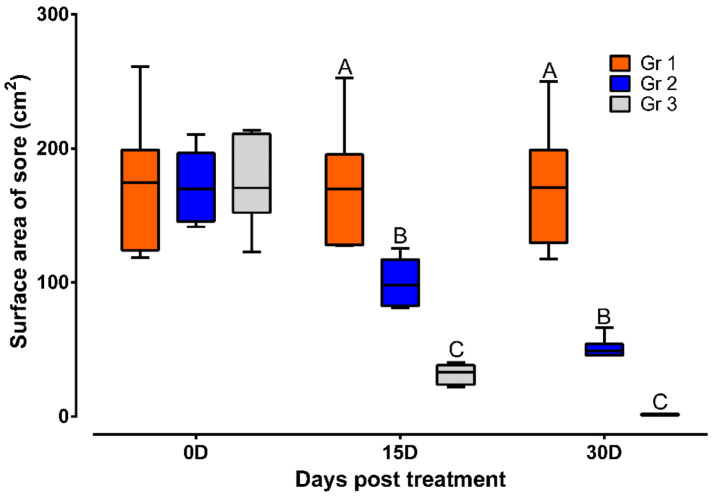
Sore surface area in different treatment groups. Data are shown as box plots. ^A,B,C^ Value with different superscript in a particular time point indicate significant difference among groups. Gr 1: infected animals without treatment, Gr 2: treatment group with ivermectin alone, Gr 3: treatment group with tri-model therapy.

**Figure 3 vetsci-08-00027-f003:**
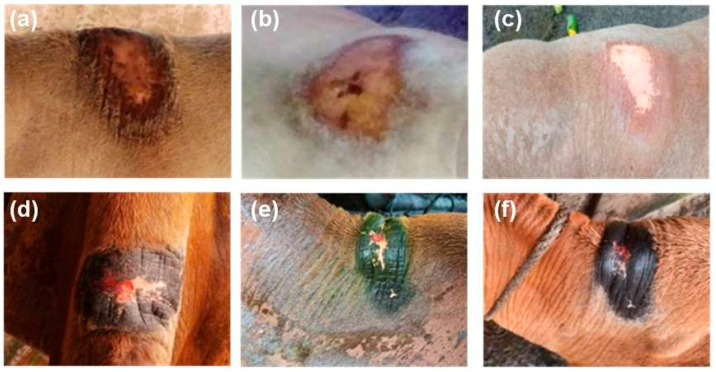
Effect of tri-model therapy on humpsore lesions, (**a**,**d**) Animals before treatment, (**b**,**e**) 15 days post-treatment, (**c**,**f**) 30 days post-treatment.

**Figure 4 vetsci-08-00027-f004:**
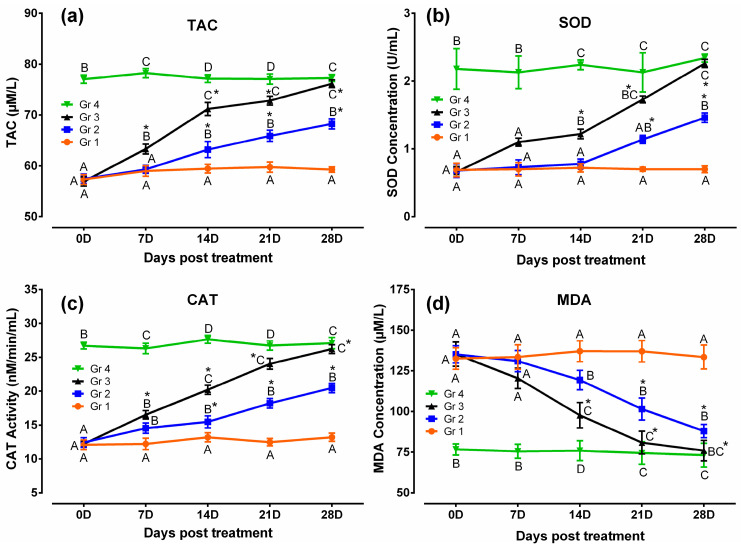
Antioxidant and oxidative stress profiles in humpsore-treated cattle: (**a**) Total antioxidant capacity, (**b**) superoxide dismutase, (**c**) catalase, and (**d**) malondialdehyde. Data are shown as mean ± SEM. Two-way analysis of variance (ANOVA) followed by Tukey post-test was performed to find out significant difference among different groups at a time and among different time points in a group. ^A,B,C,D^ Value with different superscript in a particular time point indicate significant difference among groups. Value with star (*) in a group indicates significant difference as compared to its respective 0 day value. Gr 1: infected animals without treatment, Gr 2: treatment group with ivermectin alone, Gr 3: treatment group with tri-model therapy, and Gr 4: negative control.

**Figure 5 vetsci-08-00027-f005:**
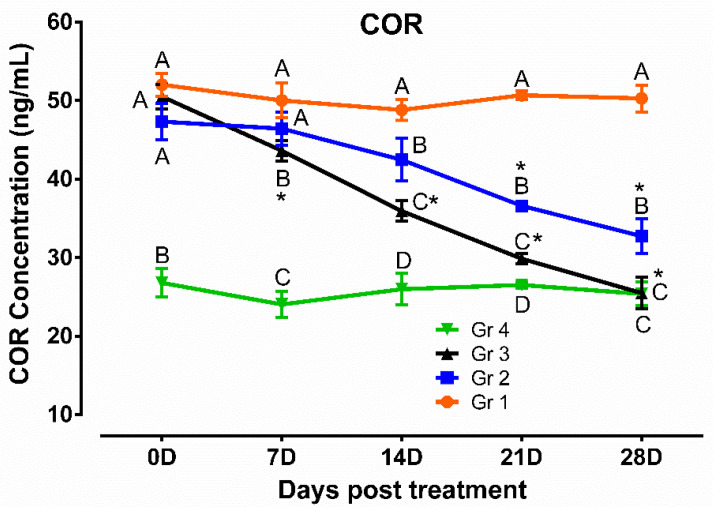
Cortisol profile in humpsore-treated cattle. Data are shown as mean ± SEM. Two-way analysis of variance (ANOVA) followed by Tukey post-test was performed to find out significant difference among different groups at a time and among different time points in a group. ^A,B,C,D^ Value with different superscript in a particular time point indicate significant difference among groups. Value with star (*) in a group indicates significant difference as compared to its respective 0 day value. Gr 1: infected animals without treatment, Gr 2: treatment group with ivermectin alone, Gr 3: treatment group with tri-model therapy, and Gr 4: negative control.

**Figure 6 vetsci-08-00027-f006:**
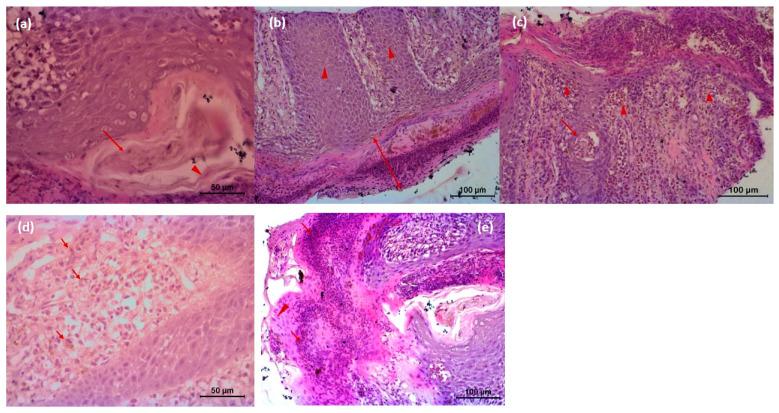
Histopathology of skin of affected animal (**a**) Microfilarial organisms (arrow) are beneath the epidermis and are surrounded by some fibrocytes (arrow head) and infiltrated cells followed by squamous epithelial cells; (**b**) microscopic lesion of parakeratotic hyperkeratosis (thickening of *Stratum corneum* of the epidermis) (arrow); keratinocytes are nucleated, and *Stratum spinosum* and *Stratum granulosum* both are proliferated. Rete pegs are elongated and hyperplastic, whichled to acanthosis (arrow head); (**c**) infiltrations of polymorphonuclear cells (eosinophils) along with some monomorphonuclear cells (arrow head). Infiltrated cell forms microabscess containing debris within (arrow); (**d**) Dermis shows extensive reaction that caused dermatitis and extensive infiltration of eosinophil and neutrophil along with lymphocytes, and eosinophilic granules (arrow) are also present in the dermis area; (**e**) severe infiltration of inflammatory cells in the *Stratum corneum* and *Stratum spinosum* (arrow). Keratinocytes retain nucleolus, highly proliferative in nature (arrow head). Cells of *Stratum spinosum* also show proliferation and contain microfilarial organism. There are exudates within epidermis between two layers. Microcavities filled with tissue debris, and inflammatory cells are observed.

**Figure 7 vetsci-08-00027-f007:**
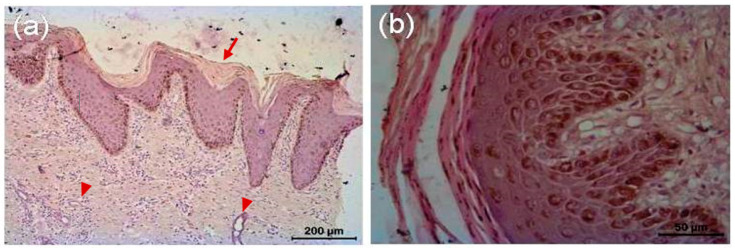
Histopathology of skin of tri-model therapy-treated animal. (**a**) Skin of treated animal shows epidermis and dermis with its different layers, almost normal *Stratum corneum*; devoid of its nucleus (arrow), *Stratum spinosum* and *granulosum* are well appreciated. Sweat glands (arrow head) are evident in the region between papillary and reticular layer of dermis. There is no infiltration. Adnexal structure is present in the biopsy material. (**b**) Different layers of epidermis after treatment.

## Data Availability

The data presented in this study are available on request from the corresponding author. The data are not publicly available due to institutional privacy policy.

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
