# Peer review of "Tri-Model Therapy: Combining Macrocyclic Lactone, Piperazine Derivative and Herbal Preparation in Treating Humpsore in Cattle"

_vetsci, 2021, doi:10.3390/vetsci8020027_

Round 1
Reviewer 1 Report
It is unfortunate that such topics, i.e. the Humpsore disease in cattle, are not often reported on in the scientific literature. However, it is an important veterinary filarial infection and can give indications for other diseases as well in line with the one health approach.
However, I have serious concerns about the design of this experiment. The authors have used 6 animals per group ( 6 infected and left untreated, 6 infected and treated with a triple therapy, 6 uninfected). Whereas this is a design to investigate the changes of a disease status returning back to normal, it may not fit to investigate the effectiveness of a new treatment regimen (triple therapy in this case) over standard medication (or the one mostly use if a standard treatment does not exist). For this, the infected and treated group should have been compared to the standard regimen. Furthermore, this is needed to show that under the same experimental condition such a treatment is more effective than the standard one. Harmonization as much as possible is a key requirement to draw conclusions appropriately. At best such treatment would be compared to the single components of a triple combination to dissect the active regimen. As sometimes this is practically not easy to perform, at least a comparison to historical data should be done, albeit given the constraints that come with it.
Other comments:
Material and Methods:
- It is not clear how animals were included into the study. How was Humpsore disease verified? No information is given on microfilarial loads. This could have been an additional indicator also to follow up treatment efficacy. This section needs more explanation, as some of the readers may not be veterinary practinioners.
- A more standardized method to evaluate the wound healing would have been an asset. For example, the diameter of the sores could have been assessed.
Results:
- The quality of the histological pictures and in particular their description is poor. At least indicators to help the inexperienced reader, who may not be a dermatologist, should be guided through the presentation.
- The graphs lack the SD.
- Mf levels over time would have been an asset
- What is the reoccurrence rate in this disease in principal? It is unclear whether parasites were cleared or may resurrect later. A longer follow up would be helpful.
Discussion:
- The discussion is difficult to follow and should be more linked to a clear hypothesis. As mentioned above, the design chosen is not optimal and it should be clearly stated, what the authors are trying to answer
- The discussion begins with a listing of the drugs used, but it should indicate more the disease itself, what is known in cattle and what drugs have been used with what results. Have for example, drugs such as melarsomine that have been proven effective in cattle against filarial infections been used? More information would help to put the data obtained into perspective.
- In the discussion, the authors are talking about using Papaya Latex and its beneficial effects in this study. However in the material and method section, they stated to use Himax ointment. This is not clear, as per content its not the same.
Author Response
Comments and Suggestions for Authors
It is unfortunate that such topics, i.e. the Humpsore disease in cattle, are not often reported on in the scientific literature. However, it is an important veterinary filarial infection and can give indications for other diseases as well in line with the one health approach.
Response: We thank the reviewer for his/her enthusiasm for our study.
However, I have serious concerns about the design of this experiment. The authors have used 6 animals per group ( 6 infected and left untreated, 6 infected and treated with a triple therapy, 6 uninfected). Whereas this is a design to investigate the changes of a disease status returning back to normal, it may not fit to investigate the effectiveness of a new treatment regimen (triple therapy in this case) over standard medication (or the one mostly use if a standard treatment does not exist). For this, the infected and treated group should have been compared to the standard regimen. Furthermore, this is needed to show that under the same experimental condition such a treatment is more effective than the standard one. Harmonization as much as possible is a key requirement to draw conclusions appropriately. At best such treatment would be compared to the single components of a triple combination to dissect the active regimen. As sometimes this is practically not easy to perform, at least a comparison to historical data should be done, albeit given the constraints that come with it.
Response: The authors agree with the concerns regarding the experimental design of the experiment. Actually, we had one group which was treated with ivermectin alone but didn't include that in the manuscript. As suggested, we have included that group in the revised manuscript and compared our tri-model therapy with standard treatment (ivermectin alone) and our treatment therapy recorded improvement over the established treatment. Accordingly, whole manuscript has been modified.
Other comments:
Material and Methods:
- It is not clear how animals were included into the study. How was Humpsore disease verified? No information is given on microfilarial loads. This could have been an additional indicator also to follow up treatment efficacy. This section needs more explanation, as some of the readers may not be veterinary practinioners.
Response: The animals were selected based on clinical humpsore lesions. Animals with a single circular or irregular shaped large/chronic wound (>12 cm diameter) were selected for the current experiment. Humpsore was verified by detection of microfilaria in skin scraping (Figure 1) of the infected animals. We did not quantify the Mf load in the animals which we justify as follows.
Obviously there are techniques to quantify the microfilaria in the skin lesion for another filarial worm, Onchocerca volvulus (Scheider et al., 1976). But this method is applicable for O. volvulus because the adult parasitic stages are seen in the subcutaneous tissue and microfilaria remains in dermis (Eberhard et al., 2010). As a result, this parasite does not produce superficial lesions. On the contrary, Stephanofilaria lives in epithelial layers of epidermis. Subsequently, the parasite causes inflammatory changes of rete Malpighii and there is destruction of hair follicles and skin glands. As the infection progresses superficial wounds develop. But severity of lesion is not dependent on quantum of parasite but it varies due to secondary bacterial infection or season (Soulsby, 1982). Therefore, in our opinion examination of skin scraping and quantification of larvae/adult will not be an indicator for correlation of skin lesion and worm load unlike O. volvulus. For that reason, for confirmatory diagnosis, we have demonstrated the causative organism from the skin lesion of infected animal.
Scheiber P, Braun-Munzinger RA, Southgate BA. A new technique for the determination of microfilarial densities in onchocerciasis. Bull World Health Organ. 1976;53(1):130-3.
Eberhard ML, Ruiz-Tiben E, Korkor AS, Roy SL, Downs P. Emergence of Onchocerca volvulus from skin mimicking Dracunculiasismedinensis. Am J Trop Med Hyg. 2010 Dec;83(6):1348-51. doi: 10.4269/ajtmh.2010.10-0475. PMID: 21118947; PMCID: PMC2990057.
Soulsby, E.J.L. (1982) in: Helminths , Arthropods and Protozoa of Domesticated Animals. BailliereTindall, 110 Greycoat Place, London SW1P 1SB. Pp 321.
- A more standardized method to evaluate the wound healing would have been an asset. For example, the diameter of the sores could have been assessed.
Response: The authors agree with the reviewer. The surface area of the wound has been assessed based on length and width of the wound. This has been included in the revised manuscript.
Results:
- The quality of the histological pictures and in particular their description is poor. At least indicators to help the inexperienced reader, who may not be a dermatologist, should be guided through the presentation.
Response: As suggested, we have included indicators in the histopathology images and result section.
- The graphs lack the SD.
Response: WE have included SEM in the graphs.
- Mf levels over time would have been an asset
Response: We did not measure Mf levels over time in our present study. Justification has already been provided above.
- What is the reoccurrence rate in this disease in principal? It is unclear whether parasites were cleared or may resurrect later. A longer follow up would be helpful.
Response: Recurrence of the disease was not detected in any of the treated animals (tri-model therapy group) for one year. However, further study is needed to record the recurrence rate over a longer time interval.
Discussion:
- The discussion is difficult to follow and should be more linked to a clear hypothesis. As mentioned above, the design chosen is not optimal and it should be clearly stated, what the authors are trying to answer
Response: The discussion section has been reshaped as suggested by the reviewer.
- The discussion begins with a listing of the drugs used, but it should indicate more the disease itself, what is known in cattle and what drugs have been used with what results. Have for example, drugs such as melarsomine that have been proven effective in cattle against filarial infections been used? More information would help to put the data obtained into perspective.
Response: The discussion section has been reshaped as suggested by the reviewer. The points suggested by the reviewer has been taken care of.
- In the discussion, the authors are talking about using Papaya Latex and its beneficial effects in this study. However in the material and method section, they stated to use Himax ointment. This is not clear, as per content its not the same.
Response: That portion has been deleted.

Reviewer 2 Report
Dear authors,
This paper shows the good results obtained to treatment stephanofilariasis in cattle with ivermectin, piperazine derivative and herbal preparation. The manuscript is well structured and written. However, the treatment with ivermectin has already been shown to be effective for this parasite. As the authors indicate in line 59, there are already studies where the effectiveness of ivermectin has been demonstrated, so the results of this work do not present scientific novelty. For this reason, I recommend that the authors include a fourth study group treated with one of the existing treatments, to verify that the treatment they propose represents an improvement compared to the already established treatments. Besides, other changes should be made in the manuscript:
- Line 33: change “Hisopatholgical” to “Histopathological”.
- Line 47: change “markrt” to “market”.
- Line 66: change “syudy” to “study”.
- Line 89: the hours must be correctly written.
- Line 119: before carrying out a parametric test, such as an ANOVA, the authors should ensure that their data present a normal distribution by carrying out a normality test and, also, a homoscedasticity test.
- Lines 153 and 160: eliminate “****denotes p<0.0001”. According to the statistical analysis, the p-value < 0.05 has been considered to carry it out. Therefore, a smaller p-value is not necessary to indicate it, since it does not give additional information. Lower p-values do not mean greater significance.
- Line 254: change “Avermectin” to “ivermectin”.
Author Response
Dear authors,
This paper shows the good results obtained to treatment stephanofilariasis in cattle with ivermectin, piperazine derivative and herbal preparation. The manuscript is well structured and written. However, the treatment with ivermectin has already been shown to be effective for this parasite. As the authors indicate in line 59, there are already studies where the effectiveness of ivermectin has been demonstrated, so the results of this work do not present scientific novelty. For this reason, I recommend that the authors include a fourth study group treated with one of the existing treatments, to verify that the treatment they propose represents an improvement compared to the already established treatments. Besides, other changes should be made in the manuscript:
Response: We thank the reviewer for his/her enthusiasm for our study. The authors agree with the concerns regarding the experimental design of the experiment. Actually, we had one group which was treated with ivermectin alone but didn't include that in the manuscript. As suggested, we have included that group in the revised manuscript and compared our tri-model therapy with standard treatment (ivermectin alone) and our treatment therapy recorded improvement over the established treatment. Accordingly, whole manuscript has been modified.
- Line 33: change “Hisopatholgical” to “Histopathological”.
Response: It has been changed
- Line 47: change “markrt” to “market”.
Response: It has been changed
- Line 66: change “syudy” to “study”.
Response: It has been changed
- Line 89: the hours must be correctly written.
Response: Modification has been made
- Line 119: before carrying out a parametric test, such as an ANOVA, the authors should ensure that their data present a normal distribution by carrying out a normality test and, also, a homoscedasticity test.
Response: Normality test and a homoscedasticity test have been performed and included in the revised manuscript.
- Lines 153 and 160: eliminate “****denotes p<0.0001”. According to the statistical analysis, the p-value < 0.05 has been considered to carry it out. Therefore, a smaller p-value is not necessary to indicate it, since it does not give additional information. Lower p-values do not mean greater significance.
Response: the authors agree with the reviewer and accordingly, necessary modifications have been made in the revised manuscript.
- Line 254: change “Avermectin” to “ivermectin”:
Response: It has been changed

Round 2
Reviewer 1 Report
Improvements have been made and improved the manuscript.
Reviewer 2 Report
Dear authors,
The manuscript has has improved substantially. However, a modification must be made:
- Line 35: change "Histopatholgical" to "histopathological".
